

# The 2014 coral bleaching and freshwater flood events in Kāneʻohe Bay, Hawaiʻi

Keisha D. Bahr, Paul L. Jokiel and Kuʻulei S. Rodgers

Hawaiʻi Institute of Marine Biology, University of Hawaiʻi, Kāneʻohe, HI, USA

## ABSTRACT

Until recently, subtropical Hawaiʻi escaped the major bleaching events that have devastated many tropical regions, but the continued increases in global long-term mean temperatures and the apparent ending of the Pacific Decadal Oscillation (PDO) cool phase have increased the risk of bleaching events. Climate models and observations predict that bleaching in Hawaiʻi will occur with increasing frequency and increasing severity over future decades. A freshwater "kill" event occurred during July 2014 in the northern part of Kāneʻohe Bay that reduced coral cover by 22.5% in the area directly impacted by flooding. A subsequent major bleaching event during September 2014 caused extensive coral bleaching and mortality throughout the bay and further reduced coral cover in the freshwater kill area by 60.0%. The high temperature bleaching event only caused a 1.0% reduction in live coral throughout the portion of the bay not directly impacted by the freshwater event. Thus, the combined impact of the low salinity event and the thermal bleaching event appears to be more than simply additive. The temperature regime during the September 2014 bleaching event was analogous in duration and intensity to that of the large bleaching event that occurred previously during August 1996, but resulted in a much larger area of bleaching and coral mortality. Apparently seasonal timing as well as duration and magnitude of heating is important. Coral spawning in the dominant coral species occurs early in the summer, so reservoirs of stored lipid in the corals had been depleted by spawning prior to the September 2014 event. Warm months above 27 °C result in lower coral growth and presumably could further decrease lipid reserves, leading to a bleaching event that was more severe than would have happened if the high temperatures occurred earlier in the summer. Hawaiian reef corals decrease skeletal growth at temperatures above 27 °C, so perhaps the "stress period" actually started long before the bleaching threshold of 29 °C was reached. Hawaiʻi is directly influenced by the PDO which may become a factor influencing bleaching events in subtropical Hawaiʻi in much the same manner as variations in the El Niño Southern Oscillation (ENSO) influences bleaching events at low latitudes in the tropical Pacific. Records show that offshore temperatures measured by satellite will not always predict inshore bleaching because other factors (high cloud cover, high wind and wave action, tidal exchange rate) can limit inshore heating and prevent temperatures in the bay from reaching the bleaching threshold. Low light levels due to cloud cover or high turbidity can also serve to prevent bleaching.

Corresponding author
Keisha D. Bahr, kbahr@hawaii.edu

## INTRODUCTION

Coral bleaching is a stress response that results in the degeneration and expulsion of symbiotic algae known as zooxanthellae from the coral host (*Douglas, 2003*). As a result, the white skeleton becomes visible through the transparent coral tissue giving the organism a "bleached" white appearance. Bleaching is a highly subjective term used to describe a variety of conditions pertaining to low symbiont densities or loss of photosynthetic pigments of the algal symbionts (reviewed by *Fitt et al., 2001*; *Jokiel, 2004*) and not a simple direct response to elevated sea surface temperatures (SST). Local and global abiotic (e.g., irradiance, salinity, ultraviolet radiation, etc.) and biotic (e.g., disease) stressors may act alone or in combination to cause bleaching (reviewed in *Jokiel, 2004*). The zoox-anthellae provide photosynthetic products vital to meeting host energetic requirements (*Falkowski et al., 1984*). Bleached reef corals cannot survive very long unless conditions change and the symbiosis is reestablished (*Baker, 2001*); however, in severe cases high mortality occurs among the bleached corals (e.g., *Wilkinson et al., 1999*). Chronic or widespread loss of symbionts disturbs the metabolism of the coral host and can lead to delayed or reduced reproduction, tissue degradation, reduced growth, and death of the affected tissue (*Michalek-Wagner & Willis, 2001*; *Szmant & Gassman, 1990*; *Williams & Bunkley-Williams, 1990*).

Since the 1980s, regional bleaching events have occurred on coral reefs throughout the world with increasing frequency and increasing geographic extent. The first massive bleaching event off Panama in 1983 was followed by more frequent and severe events throughout the world (*Glynn, 1991*; *Graham, 1994*). One of the largest mass bleaching events occurred in the Seychelles in 1998 where more than 90% of live coral cover was lost (*Wilkinson et al., 1999*). Another mass bleaching event in 2005 affected 80% of the coral reefs in the Caribbean and over 40% of corals died at many locations across 22 countries (*Eakin, Lough & Heron, 2009*). The occurrence and severity of mass coral bleaching has increased continuously over the last two decades. As a result, almost every reef region in the world has now suffered extensive stress or coral mortality.

These large-scale bleaching events correlate with elevated sea surface temperatures (SST), especially during El Niño Southern Oscillation (ENSO) (*Williams & Bunkley-Williams, 1990*; *Glynn, 1991*; *Wellington et al., 2001*). However, major bleaching events have also occurred outside of ENSO periods (*Brown, 1997*). The increase in SST has been shown to be the result of climate change due to anthropogenic release of carbon dioxide and other gasses (*Hoegh-Guldberg et al., 2007*).

Until recently, the isolated subtropical location of Hawai'i served as a haven from conditions that have ravaged coral reef communities in other areas of the world. The gradual rise in ocean temperature off Hawai'i shown in long-term records led *Jokiel & Coles (1990)* to predict that Hawaiian reefs were also approaching their upper thermal limits and that the first bleaching events were imminent. The first large-scale bleaching event in the Hawai'i region occurred during the late summer of 1996 and was monitored closely in Kāne'ohe Bay throughout the period of onset, bleaching and recovery (*Jokiel & Brown, 2004*). This event also impacted the northern part of the main Hawaiian Islands,
while the majority of the Northwestern Hawaiian Islands (NWHI) showed no warming or bleaching. In contrast, a second bleaching event during the summer of 2002 showed positive anomalies in the NWHI and normal conditions in the main Hawaiian Islands (*Brainard, 2002*; *Aeby et al., 2003*). In 2013, an apparent shifting of the Pacific Decadal Oscillation (PDO) out of the cool phase and into the warm phase led *Rodgers et al. (2015)* to conclude that another bleaching event was imminent.

The PDO index is an empirical orthogonal function (EOF) of monthly sea surface temperature anomalies over the North Pacific poleward of 20°N, calculated after the global mean SST has been removed (*Schneider & Cornuelle, 2005*). The monthly mean global average SST anomalies are removed to separate the Pacific pattern of variability from any "global warming" signal that may be present in the data. Monthly values since 1900 are reported at http://research.jisao.washington.edu/pdo/PDO.latest. The PDO has been described as a long-lived ENSO-like pattern of Pacific climate variability (*Zhang, Wallace & Battisti, 1997*). The PDO is a climate phenomenon of the North Pacific (as opposed to ENSO which affects the tropical Pacific). The PDO typically alternates between two phases, but remains in each phase for a significant period of time (10–40 years). The two phases of the PDO have been called warm phases (positive values) or cool phases (negative values). The global SST pattern shows the strong tendency for temperatures in Hawaï̈i and the central North Pacific to be anomalously cool when SSTs along the coast of North America are unusually warm, and vice-versa (*Graham, 1994*; *Zhang, Wallace & Battisti, 1997*; *Mantua et al., 1997*). During the past century, two major PDO eras that persisted for 20–30 years have been identified (*Mantua et al., 1997*; *Minobe, 1997*). Cool PDO regimes prevailed from 1890 to 1924 and again from 1947 to 1976, while warm PDO regimes occurred from 1925 to 1946 and from 1977 through the mid-1990's.

Coral reefs are vulnerable to storm flooding events that reduce salinity in shallow waters (*Banner, 1968*; *Jokiel et al., 1993*). Flash floods are intense, short in duration and common in Hawaï̈i. The humid air usually contains enough water to produce heavy rains at any time, but normally flash floods are associated with upper-level forcing (*Jokiel, 2006*). Flooding often occurs when convective cells are formed or enhanced by orographic effects and become anchored against high-vertical relief features. These conditions occurred during the night of 19 July 2014 in the Koʻolau Mountains at the head of the Waiʻāhole–Waikāne valley watershed. The resulting flash floodwaters drained onto inshore reefs in the northern part of Kāneʻohe Bay. Reef corals can live under natural conditions at salinity ranging from 25 to 42‰ (*Coles & Jokiel, 1992*). Reduction in salinity to 15–20‰ for a 24 h period or longer produces a lethal osmotic environment for the corals and their endosymbionts (*Mayfield & Gates, 2007*). Such low salinities result in extensive bleaching and high mortality (*Coles & Jokiel, 1992*; *Jokiel et al., 1993*). Freshwater flood events have been documented in Kāneʻohe Bay, Hawaï̈i during May 1965 (*Banner, 1968*), again during 1988 (*Jokiel et al., 1993*) and recently the more localized event occurred during flash floods in July 2014 (*Bahr, Rodgers & Jokiel, 2015*) for a frequency of re-occurrence of approximately 25 years. During the 1965 flood, the freshwater discharged into the bay in a 24 h period was calculated to be equivalent to a surface layer of 27 cm over the entire

bay (*Banner, 1968*). The reduction in salinity in surface waters caused massive mortality of coral reef organisms, with near total mortality of corals to a depth of 1–2 m in the inshore regions. Twenty-three years later, a comparable storm flood resulted in similar large-scale destruction of the reef flat corals in shallow (<2 m) water (*Jokiel et al., 1993*). Data on the frequency and intensity of these events are important in the understanding of long-term trends in coral reef ecology (*Rodgers et al., 2015*).

The 2014 flash flood and subsequent high temperature bleaching event provided an opportunity to test the hypothesis that high temperature and low salinity have a synergistic negative effect on corals (*Coles & Jokiel, 1978*). In addition, this high temperature bleaching event in relation to a long-term temperature record taken offshore is relevant to hind cast information as well as projections that predict more severe and more frequent occurrences over time (*Jokiel & Brown, 2004*; *Buddemeier et al., 2008*; *Hoeke et al., 2011*). The first reports of bleaching led us to the hypothesis that the 2014 bleaching event would be more severe than the previous 1996 event in Kāneʻohe Bay, and would have greater impact in the area previously damaged by the storm flood.

## METHODS

### Study site description

Kāneʻohe Bay, located on the northeast coast of Oʻahu, Hawaiʻi (21°, 28′N; 157°48′W), is a shallow (>50% of the bay is less than 3.3 m) embayment and receives drainage from nine watersheds. It is the largest sheltered body of water in the eight main Hawaiian Islands with a total surface area of 41.4 km² at mean surface levels (*Jokiel, 1991*; *Bahr, Jokiel & Toonen, 2015*). The bay contains numerous patch reefs and is bounded by a barrier reef on the seaward side creating restricted water exchange with the open ocean (Fig. 1). Also, most of the shoreline is ringed by shallow fringing reef 0.3–0.9 m deep (*Jokiel, 1991*; *Bahr, Jokiel & Toonen, 2015*)). Due to restricted circulation in this bay the temperatures are historically 1–2 °C higher than the open ocean in the summer months. Consequently, the corals in the bay are already living at temperatures that offshore ocean reefs will not experience for many years under various scenarios of global warming (*Buddemeier et al., 2008*; *Bahr, Jokiel & Toonen, 2015*).

Hawaiian waters contain about 40 coral species from eight families, of which only a few species are abundant (*Maragos, 1972*; *Bahr, Jokiel & Toonen, 2015*). Circulation patterns and environmental conditions control the distribution and abundance of coral species in the bay (*Smith et al., 1981*); however, historical anthropogenic influences (e.g., nutrient enrichment, dredging, and urbanization) have reduced coral coverage in the south bay (detailed in *Bahr, Jokiel & Toonen, 2015*). Corals in Kāneʻohe Bay are most commonly found on the reef crests and slopes of the fringing and patch reefs (summarized in *Smith, Chave & Kam, 1973*; *Jokiel, 1991*). The seaward portion of the barrier reef contains a lower percentage of live coral cover averaging about 5–10%, most of which are high wave energy coral species (e.g., *Pocillopora meandrina* and *Porites lobata*). Coral cover in the landward lagoon waters ranges between 30 and 90%, which is among the highest in the main Hawaiian Islands (*Smith et al., 1981*; *Rodgers et al., 2015*). The dominant corals

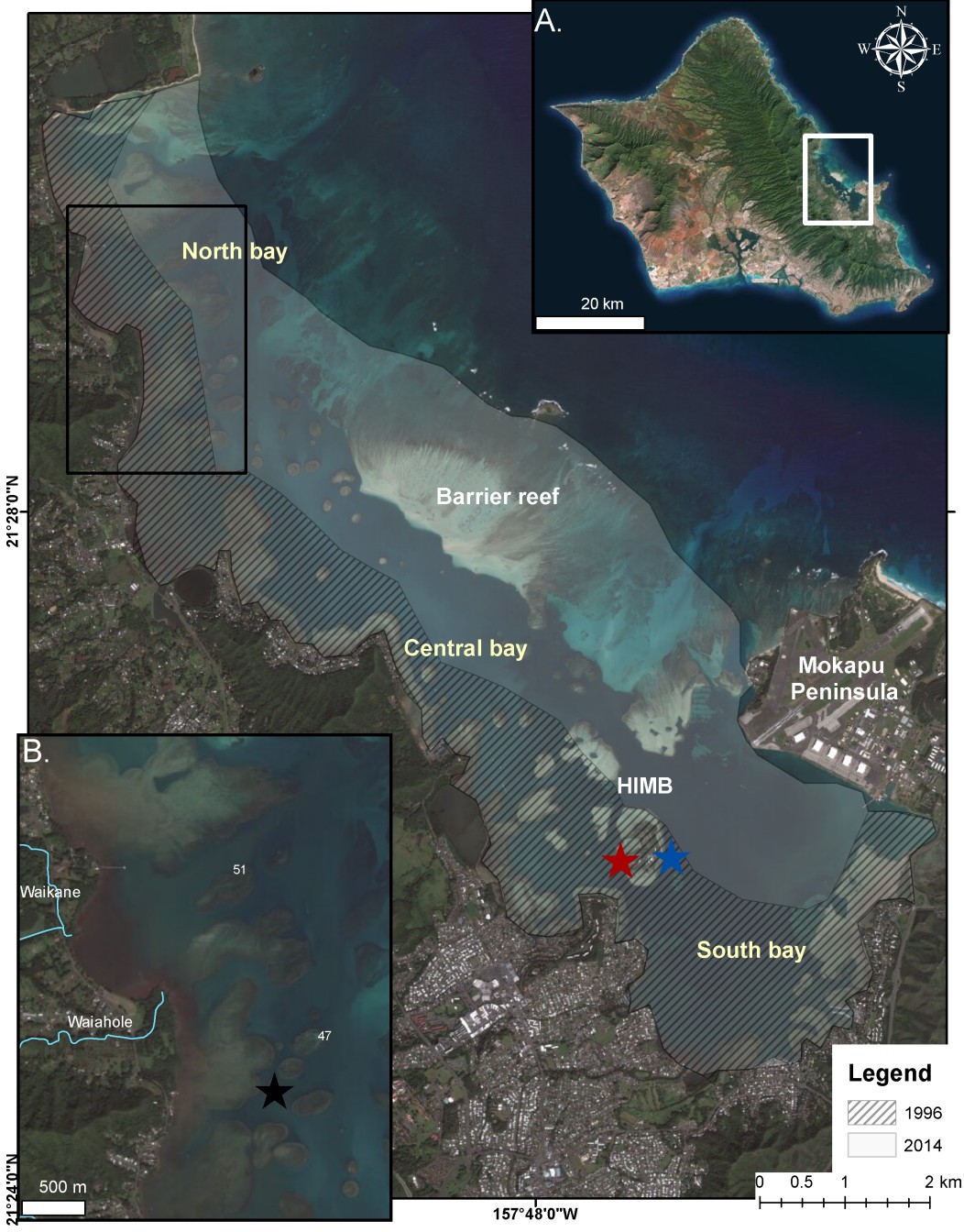

**Figure 1  Bleaching events in Kāneʻohe Bay.** Extent of impact in the 1996 (cross-hatch) and 2014 (gray) bleaching events in Kāneʻohe Bay, Oʻahu Hawaiʻi. (A) Enlarged photograph of Oʻahu, Hawaiʻi. Kāneʻohe Bay is indicated by the white square. (B) Impacted area from the proceeding freshwater event (July 2014) is indicated in the area bounded by the black square. Location of temperature recorders are shown on reef flat (black star), back reef (red star) and on reef slope (blue star).

*Montipora capitata* and the Hawaiian endemic species, *Porites compressa*, comprise ~90% of the coral coverage in Kāneʻohe Bay (*Edmondson, 1929*; *Maragos, 1972*; *Jokiel, 1991*). Other abundant species include *Pocillopora damicornis*, *Leptastrea purpurea*, *Porites lobata*, and *Fungia scutaria*. A total of 22 species of coral are now protected under the Endangered Species Act, but none are found in Hawaii (*NOAA, 2014*). Surveys and assessments of this research were conducted under the Hawaiʻi Institute of Marine Biology Special Acitivity Permit 2015–7.

## Surveys

### Freshwater kill

The freshwater event occurred during the night of 19 July 2014. The extent of the area damaged by reduced salinity was determined on 25 July 2014 based on observations of dead organisms as well as salinity measurements (YSI, model number 85-25FT) (Video S1) among the northern leeward patch and fringing reefs in close proximity to stream mouths (Fig. 1). The initial quantitative impact assessment of the freshwater event was conducted on 20 August 2014. On patch reef 37, five 25 m transects were strategically placed to include the reef flat on the north, south, east, and west facing sides of the reef in addition to a central transect deployed parallel to shore (Fig. 1). GPS locations were taken at the beginning and end of each transect using a Garmin GPSMAP 76. These sites were also temporarily marked with rebar and color-coded cable ties to allow for transect identification during resurveys. Benthic coral cover was determined using high resolution digital images taken along each 25 m transect using a Canon G12 zoom digital camera with an Canon WP-DC34 underwater housing. The camera was mounted to an aluminum monopod frame, 73 cm from the substrate to provide a 50 cm × 69 cm image. The 50 high resolution non-overlapping images from each 25 m transect were imported into an ecological analysis program PhotoGrid (*Bird, 2001*) where 25 randomly selected points were projected onto each image for a total of 1,250 points per transect. Relative abundance and composition of benthic organisms and substrate were quantified including individual coral species, higher taxonomic algal functional groups (e.g., coralline algae, turf, macro, etc.) and abiotic substrate. Coral recovery surveys were conducted on 20 February 2015 on the same transects.

### Bleaching event

Broad scale surveys were conducted on 14 October 2014 to determine the severity, extent, and spatial patterns of bleaching on the fringing, patch, and barrier reefs in Kāneʻohe Bay ($n = 144$ sites). Rapid assessments of coral cover by species and condition at all sites were conducted by a vessel operator and tender (KS Rodgers), navigator and data recorder (KD Bahr) and surveyor (PL Jokiel). The visual estimate technique was conducted in clear calm conditions from the surface through a look box and/or snorkeling along the reef edge and across reef flats within a 5 min observation period generally covering a swath of 25 m in length and 4 m in width. This method has been evaluated and quantitatively compared to eight other methods (i.e., quadrat, random, point intercept transect, CRAMP RAT, video transect, towed-diver, photographic transect, NOAA ground truth) and produces results

comparable to techniques that require much more effort (*Jokiel et al., 2015*). The visual estimate technique is very rapid (5 minutes per transect) but consequently shows higher variance and requires an observer with extensive experience in quantitative assessments of coral cover and condition (*Jokiel et al., 2015*). Through this technique, we were able to evaluated bleaching severity and extent across the large sampling area and reduce variance by using the same experienced observer (PL Jokiel) at all sites. Moreover, the large number of sites increases confidence in the results. Coral condition was classified visually as "bleached" (pure white), "pale" (obvious pigment loss but some color), "normal" and newly "dead" skeleton using the same methodology and classification originally established by *Jokiel & Coles (1974)*. Estimates were made in the areas of high coral cover at a depth of 1–2 m. Subsequently, coral recovery surveys occurred on 1 December 2014 to assess coral mortality and pigmentation on the same reefs throughout the bay ($n = 120$ sites) using the same methods and observer.

## Analysis of historical sea water temperature (SST) and meteorological data

Offshore SST records based on satellite and other data at weekly intervals on a one-degree latitude–longitude grid were obtained from the Integrated Global Ocean Services System (IGOSS) at the International Research Institute for Climate and Prediction (IRI) website. These data are produced at the US National Meteorological Center (NMC) using optimum interpolation (OI) analysis of SST fields blended from ship, buoy and bias-corrected satellite data. Where satellite data are used they are adjusted for biases using the method of *Reynolds (1988)* and *Reynolds & Marsico (1993)*. A description of the OI analysis can be found in *Reynolds & Smith (1994)*. *Reynolds (1993)* gives examples of the effect of recent corrections. The IGOSS-NMC data sets were analyzed for the time period from November 1981 through December 2014. The data set for 21–22°N, 157–158°W covers all waters offshore off east Oʻahu. Weekly IGOSS-NMC temperature plots and temperature anomaly plots for the Hawaiian region (16–33°N, 150–180°W) were also analyzed (Fig. 4).

A 47-year time series of twice weekly SST taken at Koko Head on Oʻahu, Hawaiʻi was developed by the National Marine Fisheries Service (NMFS) from 1955 to 1992. The sampling site was located along the steep cliffs that extend into deep water at 21°17′N; 157°41′W. SST at this location is representative of offshore central Pacific surface waters due to deep-water exposure and strong onshore currents (*Seckel & Yong, 1977*). The IGOSS-NMC and the NMFS time series overlap between 1 November 1981 and 19 March 1992. An analysis of temperature records from the two sources during the overlap period determined the legitimacy of combining the NMFS and IGOSS-NMC data into a single record (*Jokiel & Brown, 2004*). Mean summer monthly temperature in the Hawaiian region is approximately 27 °C ± 1 °C (*Jokiel & Coles, 1977*). A 30-day exposure to temperatures of only 29–30 °C will cause extensive bleaching in Hawaiian corals (*Jokiel & Coles, 1990*).

Meteorological variables, irradiance and temperature in Kāneʻohe Bay were monitored throughout the bleaching events of 1996 and 2014 by an automatic recording weather station located on Moku o Loʻe at the Hawaiʻi Institute of Marine Biology (HIMB) (Fig. 1).

Continuous water temperature was recorded at 1–2 m depth at three locations in the bay during the 2014 bleaching event: (1) fore reef slope of the Moku o Loʻe Reef in an area of high coral cover adjacent to the weather station (Fig. 1 blue star), (2) back reef of Moku o Loʻe Reef at the NOAA National Ocean Service (Station ID 1612480, sensor E1, 1.2 m depth) in a sheltered area near a well-developed coral community (Fig. 1 red star), and (3) among corals in shallow water on "patch reef 37" in the northern part of the bay (Fig. 1 black star). Temperature at the HIMB weather station was recorded using a recording thermistor thermometer (Cole-Parmer Instrument Co., Vernon Hills, Illinois, USA). Temperatures on patch reef 37 were collected with an Onset HOBO Pendant (UA-002-08; Onset Computer Corporation, Cape Cod, Massachusetts, USA) recording at 10 min intervals.

## Statistical analysis

The long term Koko Head temperature trends between 1956 and 2014 were analyzed using a linear regression and mean sea surface temperatures were analyzed with a One way ANOVA. Assumptions of normally distributed residuals and homoscedasticity were assessed through graphical analyses of the residuals. Detailed transect surveys were analyzed using a paired $t$-test. All statistical analyses were processed using (*JMP Pro, 2014*) Pro 11 software (SAS Institute Inc., USA).

## RESULTS

### Freshwater event

Previous freshwater storm events in Kāneʻohe Bay in 1965 (*Banner, 1968*) and in 1988 (*Jokiel et al., 1993*) killed shallow water corals throughout the bay. In contrast, the event on 19 July 2014 was restricted to northern patch and fringing reefs located in close proximity to Waiʻāhole and Waikāne stream mouths (Fig. 1). Within a 24 h period, 24 cm of rainfall was measured at the Waiʻāhole rain gauge which increased the stream daily mean discharge by an order of a magnitude from 0.74 $m^3$ $s^{-1}$ to 24 $m^3$ $s^{-1}$ over a 48 h period (*USGS, 2014*). Hourly rainfall data during the event revealed the largest amount of rainfall (4.1 cm $h^{-1}$) occurred at the lowest tide exposing those shallow reef communities to further reductions in salinity (Fig. S1). Temperatures prior to the storm event ranged from 27.4 °C to 29.2 °C on adjacent patch reefs. During the flood event the input of freshwater caused temperatures on the adjacent reef flat to decrease by 1 °C and average irradiance levels deceased by 55%. These low irradiance regimes persisted for three days following the event.

Extensive bleaching and mass mortality of corals and cryptic reef dwellers (e.g., eels, crabs, shrimp) were observed on more than 50% of the leeward surveyed patch and fringing reefs following the storm (Video S1). Coral tissues were killed and sloughed off and the dead skeletons were subsequently overgrown by turf algae. Of the major benthic substrate types surveyed, an increase was shown in all substrate types (i.e., macroalgae, turf, calcareous coralline algae, and silt) with the exception of coral cover between August 2014 and February 2015 (Fig. S2). On the surveyed patch reef 37, 22% of total coral cover died as the direct result of the freshwater event.

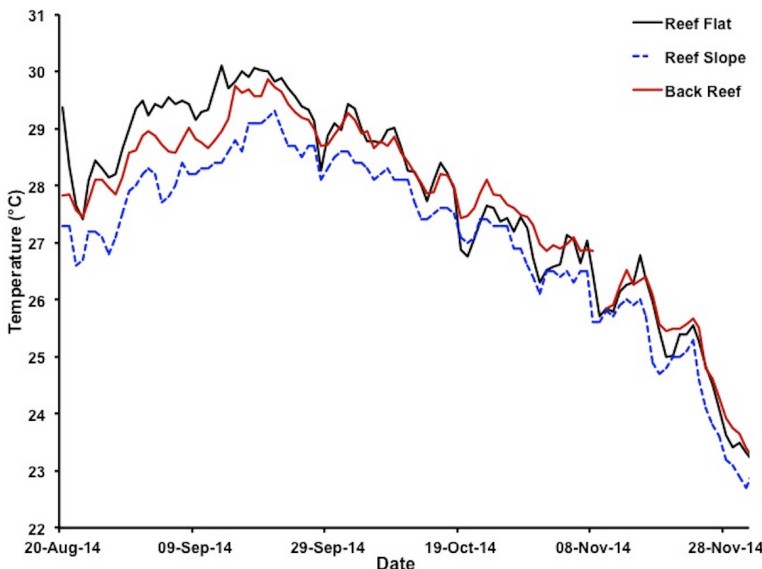

**Figure 2** **Variations in water temperatures in Kāneʻohe Bay at different reef locations.** Water temperatures at depth of less than 1 m on reef flat (solid black line for patch reef 37), depth of 1–2 m on reef slope (dashed blue line for HIMB weather station) and on the back reef at 1–2 m depth (solid red line for NOAA tide gauge station) during the 2014 bleaching event 20 August 2014–1 December 2014. Color of lines matches the color of the stars marking the respective location of each measurement site as plotted in Fig. 1.

## Bleaching event

### Broad-scale bleaching surveys

Extreme warming occurred offshore of Hawaiʻi during August–September 2014, leading NOAA (coralreefwatch.noaa.gov) to issue a bleaching warning in early September 2014 for the island of Oʻahu. Notifications gradually increased to a full bleaching alert later in the month. Reports of bleaching throughout Hawaiʻi followed. By early September mean SST in Kāneʻohe Bay reached 28.5 °C and was increasing rapidly (Fig. 2). Corals throughout the bay began to show signs of stress including contracted polyps, mucous secretion and some discoloration (Video S2). The maximum daily temperatures reached 30 °C for nearly two weeks with maximum temperatures that exceed 30 °C during mid-day on the reef flat (Fig. 2). By the middle of September, mid-day hourly water temperatures on the reef flat exceeded 31 °C. The conditions of low winds, high solar input, and increase in SST had a significant influence on the severity and extent of coral bleaching in the bay (Fig. 1).

Extensive bleaching was first recorded on 21 September 2014 (*Neilson, 2014*). Massive bleaching (80–100% of total coral cover) was observed in some areas (Fig. 3). High levels of bleaching and paling were observed throughout the bay. In the north bay, 73% of surveyed corals showed signs of bleaching and paling. Surveyed corals in the central and south bay also had high levels of bleaching and paling (62%, 70% respectively) (Fig. 3). On the barrier reef, where the total coral cover is low (5–10%), 72% of surveyed corals exhibited signs of bleaching (Fig. 3). Bleaching was observed to decrease with depth, especially in localized areas of high turbidity.

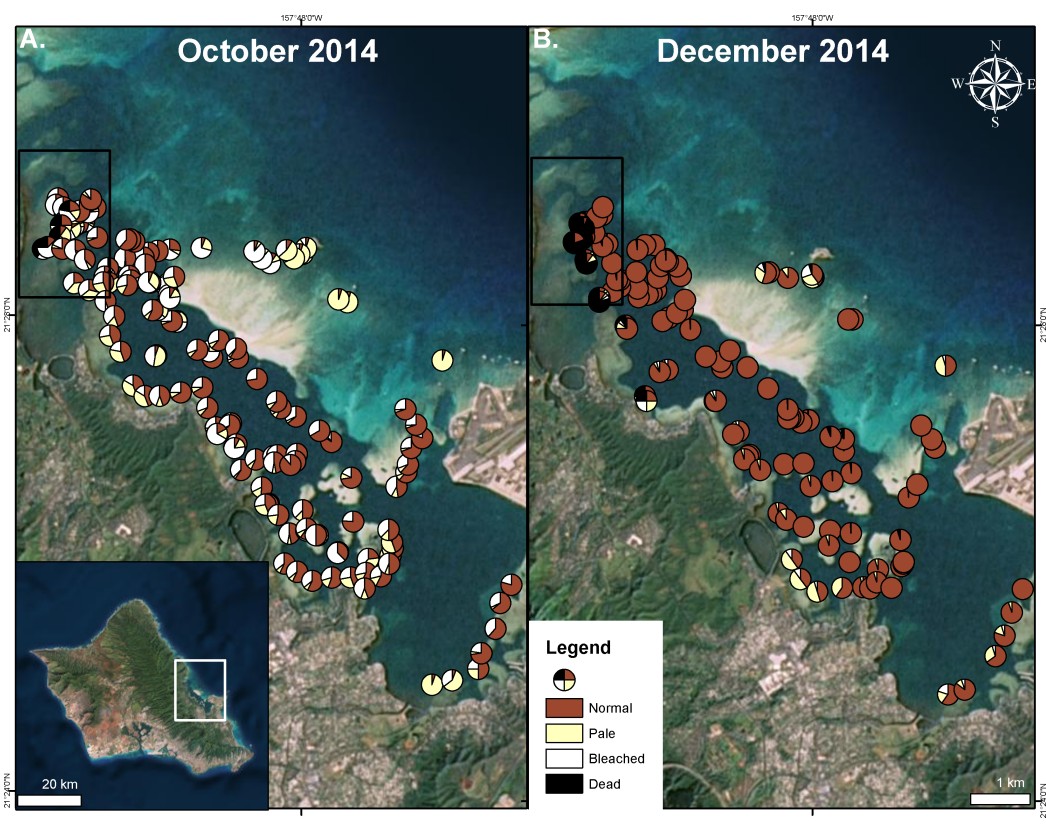

**Figure 3** **Extent of 2014 bleaching event.** Surveyed sites during the bleaching event in Kāneʻohe Bay in October 2014 (A) and after initial recovery in December 2014 (B). Area impacted by the flooding event is indicated in the black square. Proportion of surveyed corals are shown as normal (red), pale (yellow), bleached (white), and dead (black).

Relative bleaching sensitivity observed in September 2014 was similar to that reported for the August 1996 bleaching event, with members of the Pocilloporidae family as the most susceptible to bleaching. However, during the September 2014 event, colonies of *Montipora capitata* showed varied bleaching responses and were often more resistant to bleaching in comparison to the hardy *Porites compressa*. Additionally, we observed bleaching variations within species, where colonies in the same environment and often right next to each other, showed different pigmentation levels.

Recovery of pigmentation was observed in early December 2014 in most of the bay except on the barrier reef where pigment loss was still prevalent (26%) (Fig. 3). A large portion of the surveyed patch reefs were observed to recover in the north bay (75%), south bay (91%) and the central bay (89%) (Fig. 3). Coral mortality levels were estimated to be less than 2% in August 1996 (*Jokiel & Brown, 2004*), while bay wide surveys indicated average coral mortality to reach 9% following the 2014 freshwater kill and bleaching events combined (Table 1). Major mortality was largely confined to those reef areas impacted by the freshwater event, reaching nearly 60% following the bleaching event, while the reefs areas not influenced by the lowered salinity event only experienced 1% bleaching mortality (Fig. 3 and Table 1).
**Table 1 Broad-scale bleaching assessment.** Total coral cover is expressed as percent. Normal, pale, bleached, and dead are expressed as proportions of the total coral cover in the freshwater impacted areas and non-impacted areas during the bleaching assessment (14 October 2014) and the recovery assessment (1 December 2014).

| Proportion of surveyed corals | 14 October 2014: Bleaching | | | 1 December 2014: Recovery | | |
|---|---|---|---|---|---|---|
| | Impacted | Non-impacted | Overall | Impacted | Non-impacted | Overall |
| Total coral cover | 49.4 | 61 | 59.6 | 53.6 | 54.9 | 63.2 |
| Normal | 1.6 | 32.6 | 27.4 | 33.1 | 90.2 | 82.4 |
| Pale | 26.5 | 22.7 | 23 | 4.1 | 6.3 | 6 |
| Bleached | 49.4 | 44.7 | 45.3 | 2.8 | 2.6 | 2.6 |
| Dead | 22.5 | 0 | 4.3 | 60 | 0.9 | 9 |

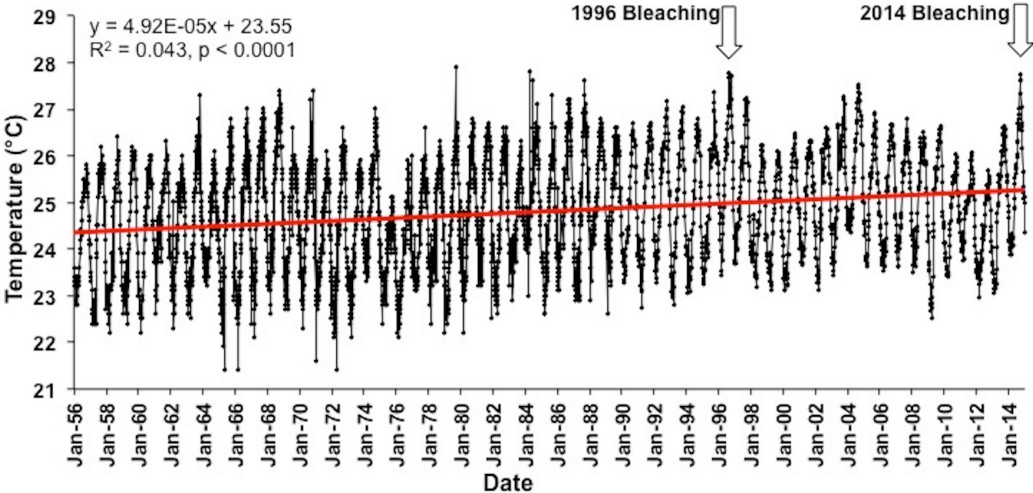

**Figure 4 Long term Sea surface temperatures (SST).** Combined SST record using IGOSS-NMC temperature data (1992–2014) and corrected NMFS data for Koko Head, Oʻahu (1956–1992).

Detailed transect surveys indicated an additional significant decline in coral cover of 29.6% between August 2014 and February 2015 (paired $t$-test; $t_{(4)} = 3.27$, $p = 0.031$). The increased temperatures in late September exacerbated mortality of those corals exposed to the preceding flooding event. Therefore, the surveyed areas exposed to the flood water experienced an overall 42% decline in coral cover between August 2014 and February 2015. The two major species *M. capitata* and *P. compressa* continued to dominate the coral cover throughout the bay as they have in the past through previous environmental perturbations since quantitative measurements have been taken (e.g., see *Smith, Chave & Kam, 1973*; *Smith et al., 1981*; *Rodgers et al., 2015*; *Bahr, Jokiel & Toonen, 2015*). We observed high mortality in certain less abundant species, such as *Pocillopora damicornis*, as also described in the 1996 bleaching event (*Jokiel & Brown, 2004*).

### Hind-casting using the HadISST data set

Offshore temperature during the 1996 bleaching event showed an offshore maximum of 27.4 °C, with one month mean temperature exceeding 27.0 °C, which might be taken as the hind cast threshold for bleaching conditions in Kāneʻohe Bay. However, if we use this value as a bleaching threshold, severe bleaching should have occurred during 1968 (maximum 28.0 °C, with 4 months exceeding 27.0 °C) and 1974 (27.8 °C with 3 months exceeding 27.0 °C). We can state with confidence that large-scale bleaching absolutely did not occur during the years 1968 and 1974. Extensive scientific observations were being made in Kāneʻohe Bay during that time by a number of scientists operating from HIMB at Moku o Loʻe with no reports of bleached corals. The classic initial research on coral bleaching was then being conducted in Kāneʻohe Bay (e.g., *Jokiel & Coles, 1974*; *Jokiel & Coles, 1977*; *Coles, Jokiel & Lewis, 1976*) so certainly any bleaching would have been detected. Likewise, there were no other reports of coral bleaching from Oʻahu or any of the Hawaiian Islands (*Jokiel & Brown, 2004*).

### Forecasting using the HadISST data set

The longest accurate temperature record available for Hawaiian waters shown in Fig. 4. A linear regression of these data (1956–2014) shows a significant positive slope ($R^2 = 0.04$, $F_{(1,4359)} = 198.2$, $p < 0.0001$) indicating a 0.02 °C increase in the annual mean temperature each year and a 1.15 °C increase over the past 58 years. Mean sea surface temperatures were significantly higher in 2014 (25.5 °C) in comparison to 1956 (24.4 °C) (One way ANOVA; $F_{(1,102)} = 30.1$; $p < 0.0001$). The trend is expected to continue as global climate change continues unabated, and the bleaching threshold will be crossed more often as time continues with increasingly long and more severe exposure to high temperature in a manner consistent with climate models developed for the region (*Buddemeier et al., 2008*; *Hoeke et al., 2011*).

## DISCUSSION

Coral reefs of the world will be increasingly vulnerable to large-scale bleaching events as the observed trend of increasing ocean temperatures continues. There is widespread agreement that global warming due to the production of anthropogenic gasses and the "greenhouse effect" is responsible for large-scale thermal bleaching throughout the world (e.g., *Pittock, 1999*; *Hoegh-Guldberg, 1999*). The year 2014 ranks as Earth's warmest since the beginning of systematic temperature recordings in 1880, according to two separate analyses by National Aeronautics and Space Administration (NASA) and National Oceanic and Atmospheric Administration (NOAA) scientists (*NASA Goddard Institute for Space Studies, 2014*). Dramatic increases have occurred in the amount of energy stored within the ocean (which is 90% or more of the total "global warming" heat) in the 1998–2014 time period (*NOAA, 2015*), which may signal another round of mass coral bleaching around the world during 2015.

This study investigated the effects of the lowered salinity and thermal bleaching events on the corals in Kāneʻohe Bay during the summer of 2014. The freshwater kill event reduced coral cover by 22.5% in the area directly impacted by flooding. The subsequent

major bleaching event during September 2014 caused extensive coral bleaching and 1.0% reduction in live coral throughout the portion of the bay not directly impacted by the freshwater event. Areas that were influenced both by the thermal and freshwater events were shown to have a 60.0% reduction in coral cover. Thus, the combined impact of the low salinity event and the thermal bleaching event appears to be more than simply additive.

## Chronology and extent of the Hawaii bleaching events

The 1996, 2002 and 2014 bleaching events in Hawaiʻi were triggered by positive summer open ocean temperature anomalies exceeding 1 °C. Moreover, periods of low winds, high solar input, and mid-day water temperatures exceeding 30 °C had a significant influence on the severity and extent of coral bleaching in Kāneʻohe Bay (Figs. 2 and 3). Observations made during the 1996 and 2014 events reinforce the role of high irradiance in accelerating bleaching of corals (*Jokiel & Brown, 2004*), suggesting that corals in high light environments may be more susceptible to bleaching (K Bahr, 2013, unpublished data).

Previous laboratory and field studies have shown Hawaiian reef corals to experience physiological stress and decrease skeletal growth above 27 °C with a bleaching threshold of 29–30 °C (*Jokiel & Coles, 1977*; *Jokiel & Coles, 1990*), so perhaps the "stress period" actually starts when seawater temperature on the reef exceeds 27 °C. Temperatures exceeded 27 °C for four weeks prior to bleaching in August 1996 and persisted an additional two weeks before dropping below 27 °C in mid-September. Another temperature peak occurred in early October 1996 prolonging but not increasing the severity of the event. These conditions persisted for an additional four weeks and subsided in mid-October. The 1996 bleaching event was restricted to the central and southern portions of the bay. In 2014, temperatures exceed 27 °C for three weeks prior to the recording of bleaching in mid-September. These temperatures persisted for five weeks and slowly dropped below 27 °C in late October (Fig. 5). Bleaching occurred over a much greater area throughout the bay in 2014 and extended to the barrier reef (Fig. 1). Higher coral mortality levels in the September 2014 thermal bleaching event are attributed to the lowered salinity event because the reefs areas not influenced by the lowered salinity event only experienced 1% bleaching mortality (Fig. 3 and Table 1). The two events were very similar in terms of magnitude and duration of thermal stress (Fig. 5), but the September 2014 event damaged a much greater area (Fig. 1). The 2014 bleaching event occurred later in the summer suggesting that seasonal timing as well as duration and magnitude of heating are important.

Coral spawning in the dominant species occurred earlier in the summer during 1996. The dominant species *Porites compressa* spawns following the new moon every month from June to September. The other dominant species *Montipora capitata* spawns after the full moon from May to September. Perhaps both species had used their reserves of lipid prior to the September 2014 bleaching event, which made them more vulnerable to the increased temperature. The subsequent warm months above 27 °C result in lower growth and presumably further decreased lipid reserves, leading to a bleaching event that was more severe than would have happened if the high temperatures occurred earlier in the season.

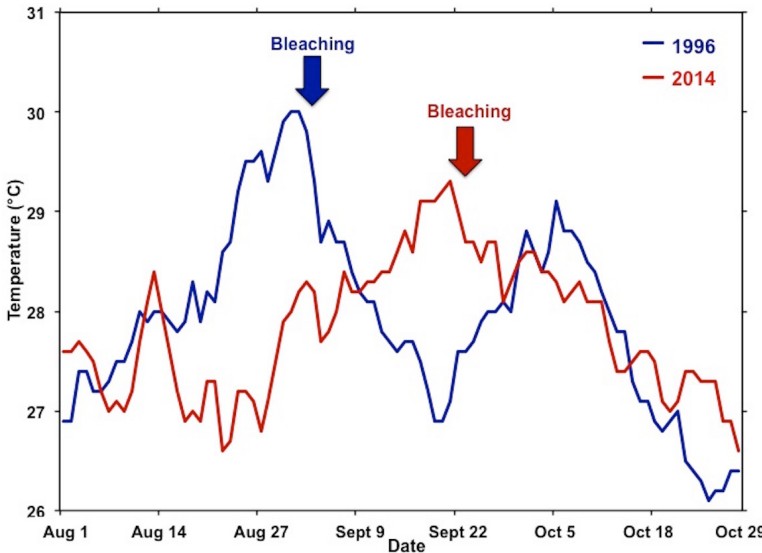

**Figure 5 Comparison of temperature during the 1996 and 2014 bleaching events.** Comparisons of the summer water temperatures during the 1996 (blue) and 2014 (red) bleaching events. Time of initial reports of bleaching for each event are indicated by respective arrow.

During the September 2014 bleaching event, several colonies of *M. capitata* were observed to be more resistant to bleaching than *P. compressa* unlike the bleaching hierarchy described by *Jokiel & Brown (2004)* during the August 1996 event. Since then, hierarchical bleaching incidence and temperature tolerances by species have been reported for Hawaiian corals by KD Bahr (2015, unpublished data) in manipulative mesocosm experiments and confirm these new observations. Also, bleaching was observed to be highly variable among individuals of the same species during the September 2014 event. These differences may be attributed to different physiological tolerance of the strains of *Symbiodinium* species and the coral hosts (*Fitt & Warner, 1995*; *Warner, Fitt & Schmidt, 1996*; *Kinzie III, Takayama & Cofforth, 2001*; *Berkelmans & Van Oppen, 2006*). Moreover, the spatial extent and patterns of bleaching responses may be influenced by factors that control the amount of solar radiation reaching the corals (i.e., cloud cover, turbidity, shading, depth) (*Kirk, 1994*; *Brown, 1997*; *Mumby et al., 2001*).

Global warming has impacted corals in various regions. *Maynard et al. (2015)* summarized available information and report that bacterial, fungal and protozoan diseases of corals, (including black band disease, yellow band disease and white syndrome) are known to have links to elevated temperature. They developed climate models that project higher temperature conditions will increase the susceptibility of corals to disease, increased pathogen abundance and virulence. Vibrios are ubiquitous in the aquatic environment with a high abundance of vibrios detected in tissues and/or organs of many marine organisms such as algae, corals, fish, and zooplankton. There are 74 species of this group are distributed among four different families (*Thompson, Iida & Swings, 2004*). *Vibrio spp.* are frequently identified as pathogens of reef corals and have been associated with bacterial

bleaching (*Kushmaro et al., 2001*; *Ben-Haim & Rosenberg, 2002*; *Weil, Smith & Gil-Agudelo, 2006*; *Rosenberg et al., 2007*). *Gregoracci et al. (2012)* studied that bacterioplankton diversity and abundance in a large tropical bay and found that higher nutrient input favored higher microbial growth including *Vibrio spp.* Thus, a rich and diverse population of vibrio is always present and nutrients brought into Kāneʻohe Bay by the flood event could have enhanced the growth of the vibrio disease during the following bleaching event. In fact, a disease outbreak in Kāneʻohe Bay was reported when the corals were beginning to recover from bleaching (*Department of Land and Natural Resources, 2015*), but this work is still in progress.

## Documenting the causes of the freshwater reef kill and predicting rate of recovery

The presence of a major flash flood was documented by rain and stream gauges. Very cold water (i.e., stream water) measured by thermographs on the reef showed low temperature at time of maximum stream flow of very short and intense duration. Dead and dying organisms were found in shallow water (<2 m) only and deeper corals were not affected due to the less dense freshwater that formed a persistent surface layer. Also, there was a strong mortality gradient with a high percentage of dead and dying reef organisms near the stream mouth grading into undamaged reefs at a distance of approximately 1 km from the stream discharge point. Nutrients and sediment input that accompanied the flood appeared to have an influence on the increase in area covered by macroalgae, turf, and silt. We can rule out high temperature as a cause because the freshwater event occurred nearly a month before the first signs of bleaching were observed throughout Hawaii. Observations on the patterns of mortality in corals and other organisms (sea cucumbers, crabs and cryptic fish such as eels) are consistent with historical evidence described in Kāneʻohe Bay from two previous freshwater kill events (*Banner, 1968*; *Jokiel et al., 1993*). Rate of recovery in the area impacted is expected to be rapid, based on previous information. Recovery was delayed after the 1965 storm event in the south basin due to the presence of two major sewer outfalls, which led to macroalga overgrowth and anoxic conditions and prevented coral recovery. After sewage discharge was terminated in 1979, these reefs recovered rapidly as shown by surveys during 1985 (*Hunter & Evans, 1995*). In contrast, the reef coral communities killed in the 1988 flood were not impacted by sewage and consequently recovered rapidly, with substantial increases in coral cover at 5 years and near complete recovery at 10 years (*Jokiel et al., 1993*). Comparison between recovery rates after the two flood events suggests that Kāneʻohe Bay coral reefs can recover quickly from natural flood disturbances, but not under chronic polluted conditions.

## Forecasting bleaching recovery

Rates of bleaching recovery in Hawaiian corals is on the order of several months as shown by field studies (*Jokiel & Coles, 1974*) and laboratory studies (*Jokiel & Coles, 1977*), but with accompanying rates of mortality depending on the extent and duration of the high temperature exposure (*Jokiel, 2004*). Recovery from the 1996 bleaching event was described by *Jokiel & Brown (2004)*. Mortality rate was very low (<2%) so the major recovery response

was from bleached condition to normal pigmentation, which occurred completely among bleached corals within 6 months. The rapid recovery of Kāneʻohe Bay bleached corals observed in this study are consistent with previous findings. Due to the low mortality levels and high recovery rates observed in August 1996, major phase shifts are not expected to occur as a result of the September 2014 bleaching event. Results of these studies suggest there is no justification for human intervention in the form of transplantation or seeding larvae. However, bleaching events in Kāneʻohe Bay are predicted to occur at much more frequent intervals with much greater mortality due to impending climate change with near elimination of Kāneʻohe Bay corals by the end of the century (*Buddemeier et al., 2008*).

## Pacific Decadal Oscillation (PDO) and bleaching events in Hawaiʻi

The long-term trend of increasing water temperature in Hawaiian waters (Fig. 4) originally led *Jokiel & Coles (1990)* to conclude that Hawaiʻi was approaching the upper bleaching threshold for corals. The first documented bleaching event in Hawaiʻi occurred in 1996 (*Jokiel & Brown, 2004*). However, there was a downward change in the temperature trend at the end of the 1997/98 ENSO (Fig. 4). Reefs in other parts of the world were severely impacted by the extreme 1998 ENSO and following events (Fig. S3), but Hawaiʻi reef escaped damage. After the 1998 event, there was a slowing of the warming trend off Hawaiʻi and what appears to be another small oscillation between the 1996 and 2014 bleaching events (Fig. 4). These changes are consistent with the findings of *Fang, Wu & Zhang (2014)*. Their model suggests that the "North Pacific Ocean decadal variability, its dominant mode (i.e., PDO), and atmospheric decadal variability, have become weaker under global warming, but with PDO shifting to a higher frequency." The mean global temperature will continue to increase as humans add additional greenhouse gasses to the atmosphere. Entering the cool phase of the PDO initially slowed predicted increases in temperature for Hawaiian coral reefs, but much warmer conditions can be expected in Hawaiian waters as the PDO reverses into the warm phase with the PDO warming increments being added to the increasing global mean. The December 2014 PDO value hit a new all-time record level of +2.51, which is the highest and hottest PDO index value since record-keeping began in 1900. The index continued to be very high into 2015 (*Joint Institute for the Study of the Atmosphere and Ocean, 2015*). Perhaps the PDO fluctuations may become a major factor influencing future mass bleaching events in Hawaiʻi, much as the patterns of ENSO influences bleaching events at lower latitudes.

## Bleaching in Hawaiʻi compared to other geographic regions

The increase of approximately 1.15 °C over the course of the 58 year record from Hawaiʻi (Fig. 4) translates into a 0.20 °C increase per decade. This value falls within the 0.07 °C–0.5 °C increase per decade range measured on various coral reefs throughout the world (summarized by *Fitt et al., 2001*). Monitoring of reefs throughout Hawaiʻi over the past 14 years (*Rodgers et al., 2015*) showed that overall coral cover and diversity in Hawaiʻi have remained relatively stable since the initial survey in 1998. During this period, coral cover on reefs in the Caribbean and other regions declined by as much as 50% due to climate change related bleaching events, increased storm damage and lowered coral growth

(*Wilkinson, 2004*). Nevertheless, Hawaiʻi is now at a threshold of more severe and frequent bleaching events projected for the near future (*Jokiel & Brown, 2004*; *Donner, 2009*; *Hoeke et al., 2011*; *Rodgers et al., 2015*).

## CONCLUSIONS

Increasing emissions of anthropogenic greenhouse gasses have resulted in global climate changes that include global warming as well as intensification of storm flood patterns. Results of this study reveal the combined impact of a lowered salinity event and the subsequent high temperature bleaching event caused a further reduction in coral cover. The influence of two of events operating in succession produced higher levels of coral mortality Kāneʻohe Bay. The temperature regimes during the September 2014 bleaching event were analogous to the August 1996 event in terms of duration and intensity, but a larger area of bleaching and coral mortality was observed. Therefore seasonal timing as well as duration and magnitude of high temperature events are important.

Until recently, subtropical Hawaiʻi escaped the major bleaching events that have devastated many tropical regions around the world, but the continued increases in global long-term mean temperatures and the apparent ending of the Pacific Decadal Oscillation (PDO) cool phase have increased the risk of bleaching events in this area. Moreover, the PDO may become a factor influencing bleaching events in subtropical Hawaiʻi in much the same manner as variations in the El Niño Southern Oscillation (ENSO) influences bleaching events at low latitudes in the tropical Pacific.

Records show that hindcasting using offshore temperatures measured by satellites will not always accurately predict inshore bleaching. It is known that other factors (cloud cover, wind and wave action, turbidity tidal exchange rate) can reduce high irradiance and limit heating in the nearshore environment and thereby prevent or reduce bleaching and mortality on coral reefs. Nonetheless, long term temperature records and documented bleaching events are consistent with the climate models and observations that predict more frequent and increasing severity of bleaching in Hawaiʻi over future decades.

## ACKNOWLEDGEMENTS

We would like to thank L Van Heukelem, C Westbrooke, C Lager, K Adams, M Bargerhuff, E Burns, E Day, K Giffen, C Guo, S Rodeghero, M Russell, J Sheu, R Sirota, J Streiffert, and A Williams for their assistance in the documentation of the freshwater and bleaching event. We would also like to thank two anonymous reviewers for their time, advice and comments which have helped improve the manuscript significantly. This is the Hawaiʻi Institute of Marine Biology (HIMB) contribution #1631 and the School of Ocean and Earth Science and Technology (SOEST) contribution #9477.

### Funding

This work was partially supported by the United States Geological Survey Pacific Coastal and Marine Science Center cooperative agreement G13AC00130. The funders had no role

in study design, data collection and analysis, decision to publish, or preparation of the manuscript.

## Grant Disclosures
The following grant information was disclosed by the authors:
United States Geological Survey Pacific Coastal.
Marine Science Center: G13AC00130.

## Competing Interests
The authors declare there are no competing interests.

## Author Contributions

- Keisha D. Bahr conceived and designed the experiments, performed the experiments, analyzed the data, wrote the paper, prepared figures and/or tables.
- Paul L. Jokiel conceived and designed the experiments, contributed reagents/materials/analysis tools, reviewed drafts of the paper.
- Kuʻulei S. Rodgers conceived and designed the experiments, performed the experiments, analyzed the data, reviewed drafts of the paper.

## Field Study Permissions
The following information was supplied relating to field study approvals (i.e., approving body and any reference numbers):

Surveys and assessments of this research were conducted under the Hawaiʻi Institute of Marine Biology Special Acitivity Permit 2015–7.

## Supplemental Information
Supplemental information for this article can be found online at http://dx.doi.org/10.7717/peerj.1136#supplemental-information.

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
