# Peer review of "The 2014 coral bleaching and freshwater flood events in Kāneʻohe Bay, Hawaiʻi"

_PeerJ, doi:10.7717/peerj.1136_

## Round 0.1 · original submission · Major Revisions

Dear Dr. Bahr et al:

Two referees have now completed their revisions. I ask you to revise your MS accordingly and make a point-by-point rebbutal letter. Please also make sure to include a ref. on additional potential causes of bleaching (e.g. infection by vibrios).
Sincerely.

Reviewer 1 ·

Basic reporting

The manuscript “The 2014 coral bleaching and freshwater reef kill events in Kāneʻohe Bay, Hawaiʻi” by Bahr at al reports two events of coral bleaching and mass mortality in Hawaiian subtropical reefs, one possibly associated with a high freshwater discharge and other associated with high seawater temperatures. The manuscript is excessively long and needs serious improvements on writing and structure. The manuscript may be accepted to publication only if authors modify it extensively.

Experimental design

Authors described the most important methodologies (surveys) at the end of “Methods” section. There is no description of how the statistical analysis were performed

Validity of the findings

Authors assumes that the observed coral bleaching and mass mortality were caused by the high freshwater discharge and high temperatures observed as truth. However I could not find in manuscript a robust approach to refute/accept this hypothesis (including statistical tests). It is not clear in introduction what were the main objectives/hypothesis/questions of the work. The manuscript must be re-written to readers follows easily this questions:
1) What were the main objectives/hypothesis/questions? (in the last paragraph of introduction section)
2) What methods authors have used to answer/test these objectives/hypothesis/questions? (be very objective in methods section)
3) What authors have found? The results section must be much clear.

In methods section (lines 200-204) authors has described how the benthic cover composition was estimated. However there is no benthic cover data on results section and there is no discussion regarding this results (only few results regarding some coral species were included in lines 292-301). In my humble opinion this would be the main result. There are many studies about coral reefs phase shifts (from coral dominated to algae dominated) and is well know that algae (specially turf algae) may induce coral disease. How benthic cover composition varies in this specific region? How about algae cover? Showing these results authors would make a rich discussion contributing much more to the coral reefs knowledge.

In results section authors showed some statistical results (e.g. line 242) but there is no description in methods of how these tests were performed.

Additional comments

Comments:

Authors must explore their benthic cover and coral bleaching incidence data better. There is no figures showing the benthic cover composition and what was the coral bleaching incidence for each coral species (at least some important species). What were the must coral species? What were the most impacted species (from bleaching and mortality)?

There is a lot of environmental data description included in results section. Maybe authors should focus on benthic cover and coral bleaching surveys data and relate these results with temperature and freshwater discharge. These relations should be done using a robust statistical approach. Authors should remove or move to Supplementary Information material the temperature temporal series figures. Would be better including benthic cover charts.

In methods section authors have included a subtopic “Forecasting and Hind-casting” subtopic however there is no forecast or hind cast results (only in discussion).

Authors cited Neilson (2014) results instead of presenting their results about coral cover and bleaching incidence. I could find the bleaching results in the provided .xlsx file however I could not find the benthic results in this file.

Please make sure to include only results from the present study in results section. There are methods and discussion in this section. For instance:

Lines 270-271: “Bleaching severity varied in the bay due to local environmental conditions (i.e., water circulation, depth).”

This is authors’ suggestion; authors must present this as such (in discussion section).

More discussion in results section (lines 282-284):
“Moreover, these observations reinforce the role of high irradiance in accelerating bleaching of corals (Jokiel and Brown, 2004), suggesting that corals in high light environments may be more susceptible to bleaching (K. Bahr, unpublished data).”

In abstract some discussion about coral spawning and coral growth were added however I could not find discussion about this topics on elsewhere in manuscript.

Suggestions:
Please remove from the text all “freshwater kills” terminology. These words denote intentionality. Please choose another term to explain the corals mass mortality followed by the high fresh water discharge. Make sure to explain (if known) the causes of this high freshwater flow to the reefs (e.g. strong rain).

Would authors include underwater pictures of the impacted/recovered reefs? This would be merely illustrative but would make readers follow their ideas better.

Include a detailed “Statistical Analysis” subtopic in methods section. Describe in details what and how statistical tests were performed make sure to include software citation.

Authors should remove the historical Hawaiian coral reef research activities and short the chronological discussion of coral bleaching events. The manuscript is too long and the results (benthic cover and coral bleaching and mortality incidence) can be discussed better.

Remove figures 5 and 6.

Include discussion about the most affected species.

Authors would include a discussion regarding coral diseases leaded by seawater warming.

Minor comments:

I could not understand this sentence (lines 215-216):
“Bleaching and recovery observations were also compiled from discussions with researchers working at other sites in Hawaiʻi.”
It sounds very unscientific.

Line 232: “negatively impacted” sounds subjective. Describe better and objectively what would it be.

Comments about the figures:
In this reviewer opinion there is too many figures in manuscript. Authors should focus on the main objective and draw effective figures to answer the raised questions in introduction.

Figure 1 – Please add the location of reefs in relation to Hawaii’s location. Readers should quickly locate Hawaii in the world as well the surveyed reefs. Figure 1B is really relevant?

Remove figures 5 and 6.


I hope my comments were useful to improve the manuscript.

Best wishes

Reviewer 2 ·

Basic reporting

Review of Keisha et al., PeerJ.

[1] K-Bay is an interesting coral reef study site for its particularities and the bulk of data accumulated, much of it, probably, thanks to the author’s contribution and efforts.
[2] The aim of the work is not clearly stated in the text. It has to be explicitated in the abstract and introduction.
[3] The structure and content of the Ms need considerable improvement. The introduction is too general, and issues not investigated are superficially mentioned. The focus of the work were the combined effects of flood (freshwater kill) and thermal bleaching over K-Bay corals in 2014 (right?). Most of the concepts and historical data on temperature in the subtropical Pacific, thermal bleaching and floods in K-Bay appear in the discussion. In other words, most of the introduction should be replaced by information presently in the discussion section. Some methods lack description (temperature and irradiance measurements) and there are results in the discussion section (some were literally repeated from Results and a few were not cited in the Results). Most of the Conclusions are in fact Discussion.
[4] Supplementary material added is not cited along the Ms.

Experimental design

Forecasting and Hind-casting: There are other methods/parameters of forecasting bleaching and coral stress than conventional temperature threshold. In fact, the hind-casting result suggests it is a weak predictor, in agreement with other studies (eg. Maynard et al., 2008. Major bleaching events can lead to increased thermal tolerance in corals. Mar Biol 155:173–182). Multivariate models take into account taxa, water flow, winds, adaptive responses and other parameters to make predictions (e.g. McClanahan et al., 2015. Regional coral responses to climate disturbances and warming is predicted by multivariate stress model and not temperature threshold metrics. Climatic Change, 1-14). Even when temperature is the main parameter considered, there are other indexes to be evaluated, such as Degree Heating Weeks (DHW), temperature variability (coefficient of variation), coefficient of temperature rate of rise. Can you improve your forecasting?

Validity of the findings

How can the authors be sure that bleaching and mortality observed after the flood were due to low salinity (freshwater kill) and not to pollutants derived from terrestrial/agricultural run-off, sewage escape or equivalent? Did you perform water quality analysis? (Chemical and microbiological).
Please clarify.

Additional comments

Specific Comments:
[1] Abstract.
Background: Besides mentioning the risk posed by thermal anomalies (past and future) in K-Bay, it should be mentioned that floods are also a risk, locally. Results: Before revealing the main findings of the work, the aim of the study should be clearly stated. Followed by a minimal description of how the authors managed to reach their objectives. Conclusions: In the abstract the conclusions are enough. Discussion items should be minimal. There is no need to split the abstract in Background, Aims, Results and Conclusions but this is the structure that should be taken into account.
[2] Introduction.
Is quite too general and confusing. It gives a definition of bleaching but returns to it later. Mentions consequences and causes of bleaching many times. Causes of bleaching mentioned are biotic, abiotic, global and local, but all mixed.
The first paragraph of the Discussion, Hawaiian coral reef research and global climate change (Discussion), much of the Chronology and extent of the Hawaii bleaching events (Discussion), Pacific Decadal Oscillation (PDO) and bleaching events in Hawaiʻi (Discussion), including Fig. 6, and Bleaching in Hawai‘i Compared to Other Geographic Regions (Discussion) are suited to ameliorate and keep the focus of the introduction (focus on K-Bay). The work is about bleaching in K-Bay. Historical data supports that major causes of bleaching in K-Bay are thermal stress and freshwater floods. Tell the readers about thermal stress (past and projections) and floods in K-Bay. The last paragraph of the introduction should contain the aims of the work and the major finding and/or conclusion.
[3] Materials and Methods.
Study site and species:
Lines 119-120: Coral cover in the landward lagoon: how much is it in %?
How many coral species are there in K-Bay? Any endemic? Any local extinction has already occurred or any endangered or vulnerable species? The authors have plenty of experience and knowledge on K-Bay reef species. Did you observe a shift of species dominance as a result of previous thermal anomalies or floods (or associated with)? This is relevant to the forecasting.
Forecasting and Hind-casting: Should be stated as part of the aims of the work (abstract and introduction). Up to this sub-title the reader did not know this was one of the aims of the study. The importance/validity of doing so, especially by the chosen method, is not uncovered anytime. Other comments in section Experimental Design.
Surveys: Even simple methods have to be described (temperature and irradiance measurements).
Bleaching survey: 144 sites were surveyed by 1 observer on 1 day (14-oct-14)? Is it correct? Again: 120 sites in 1 day and 1 observer (December 2014)? Is this correct?
[4] Results:
Try to avoid “a large percentage”and similar expressions, favoring more precise descriptions of the results.
Fig. 3: MLLW? This abbreviation was not explicated.
Lines 268-80: Is the difference mentioned significant (76% x 67%)?
Lines 282-4: This should be placed in the Discussion. Check out because it happened in other paragraphs along the Results.
Lines 299-301: Impacted and non-impacted areas (by the flood) should be distinguished in Fig. 4.
Lines 306-7: Here the decline in coral cover in the impacted area should be analyzed for the same period the analysis was performed in the non-impacted area (Aug 2014 – Feb 2015).
Neilson, 2014: this reference is not available. Part of the results were previously reported?
A table summarizing the results of bleaching in freshwater-impacted and non-impacted areas would add an overview of the major findings.
Cite the Supplementary material when appropriate, so the reader knows when to refer to it.
[5] Discussion
As mentioned before, historical data, concepts and explanation on thermal anomalies (PDO, ENSO) and floods in K-Bay should be placed in the Introduction.
Line 367-71: Part of this text is not in the Results, should be there.
Fig. 5 should be Fig. 2B, it is the same data, and should be placed in the Results section.
Lines 380-4: Results.
Hind-casting using the HadISST data set: Results.
Lines 412-5: Conclusions.
I missed a discussion on the regeneration of the reefs, since by the time of writing it was not fully achieved. How long will it take, based on past flood and thermal anomalies experience? Do the authors consider it would be useful human intervention to support regeneration in the short term? Jones and Berkelmans (2014; Flood impacts in Keppel Bay, southern great barrier reef in the aftermath of cyclonic rainfall. PLoS One. 2014 Jan 10;9(1):e84739) suggested ex-situ coral culture and transplantation and assisted recruitment to improve recovery in the short term. And in the long term? What do the authors suggest?
[6] Conclusions
Lines 491-503: Discussion

---

## Round 0.2 · Minor Revisions

Dear Dr. Bahr:

Most of the remarks of the referees have been addressed. I have just one last remark. Please add a couple of sentences in the discussion considering the possible role of
The combined/synergistic effects of lower salinity, anoxia, higher nutrient input (due to flood), and higher temperature, need to be taken into account as triggers of microbial (vibrio) growth and disease (=bleaching). Please see Line 387 – 424, line 434, line 443. Refer to http://www.ncbi.nlm.nih.gov/pubmed/15353563 and http://www.ncbi.nlm.nih.gov/pubmed/22363639.

Sincerely.

---

## Round 0.3 · accepted · Accept

Dear Authors: Congratulations on the accepted manuscript.